# Position: AI Competitions Provide the Gold Standard for Empirical Rigor in GenAI Evaluation

D. Sculley [1]  Will Cukierski [2]  Phil Culliton [2]  Sohier Dane [2]  Maggie Demkin [2]  Ryan Holbrook [2]
Addison Howard [2]  Paul Mooney [2]  Walter Reade [2]  Megan Risdal [2]  Nate Keating [2]

## Abstract

In this position paper, we observe that empirical evaluation in Generative AI is at a crisis point since traditional ML evaluation and benchmarking strategies are insufficient to meet the needs of evaluating modern GenAI models and systems. There are many reasons for this, including the fact that these models typically have nearly unbounded input and output spaces, typically do not have a well defined ground truth target, and typically exhibit strong feedback loops and prediction dependence based on context of previous model outputs. On top of these critical issues, we argue that the problems of *leakage* and *contamination* are in fact the most important and difficult issues to address for GenAI evaluations. Interestingly, the field of AI Competitions has developed effective measures and practices to combat leakage for the purpose of counteracting cheating by bad actors within a competition setting. This makes AI Competitions an especially valuable (but underutilized) resource. It is now time for the field to view AI Competitions as the gold standard for empirical rigor in GenAI evaluation, and to harness and harvest their results with according value.

## 1. Introduction

As Generative AI (GenAI) models such as Large Language Models (LLMs) become ever more important to the field and to the world, it has become clear that performing empirical evaluation of these models and methods is extremely difficult to do in a rigorous and comprehensive way. This difficulty is of course not due to lack of effort or expertise by researchers. Indeed, enormous effort and resources have been poured into creating myriad benchmarks and test cases (Chiang et al., 2024b; Fourrier et al., 2024; Hendrycks et al., 2021; Cobbe et al., 2021; Zellers et al., 2019; Chen et al., 2021b). However, even accounting for these many important efforts and achievements, our position is that **the current state of evaluation is insufficient to meet the needs of this moment in GenAI for the field and for the world.**

In our view, the root cause of this insufficiency is that the evaluation needs of GenAI models fundamentally break the paradigm of traditional benchmarking that served the field of machine learning (ML) so well during decades of remarkable progress. This breakage goes beyond the familiar difficulty of defining what, exactly, is in the training data for an LLM. **In our view, we need a broader conception of generalization for GenAI** that moves beyond the idea of generalizing to new independently drawn examples from a stationary distribution, and instead refers to performing well on tasks that are entirely novel from a model's perspective. This higher bar is rooted in commonsense standards for human intelligence (Chollet, 2019; Dennett, 1991), but has far reaching consequences, most notably that it implies the problems of *data leakage* and *contamination* in evaluation are the most pressing concerns.

Together, these factors imply that **rigorous and robust evaluation of GenAI models requires a steady source of novel tasks structured to avoid leakage, contamination, and other forms of inadvertent "cheating".** Fortunately, AI Competitions—such as those hosted on platforms like Kaggle and others—act as a solution to GenAI evaluation challenges by providing a continual source of new tasks for evaluation and significant structures to avoid leakage and related issues. We define an AI Competition as a problem or task with an objective evaluation function for ranking solutions or models in which multiple parallel attempts are made during a time-bound period by independent teams.

### 1.1. Summarizing Our Position

Our position can be summarized by the following points:

- Traditional paradigms for ML evaluation are ill-equipped to meet the demands of GenAI Evaluation.

[1]Work done at Kaggle [2]Kaggle, Inc. Correspondence to: Nate Keating <natekeating@kaggle.com>, Meg Risdal <meg@kaggle.com>.

*Proceedings of the 42nd International Conference on Machine Learning*, Vancouver, Canada. PMLR 267, 2025. Copyright 2025 by the author(s).

- Leakage should be viewed by the field as the most important pitfall to avoid in evaluations.

- GenAI evaluations should be considered leaked the moment test data has been shared online or sent over the wire to models.

- If we have to choose between reproducibility and robustness in GenAI evaluations, we should choose to prioritize robustness.

- We should replace the notion of reproducible static benchmarks with repeatable processes and procedures.

- The field should use established AI Competitions platforms as a renewable stream of novel evaluation tasks.

- The standards and practices developed that help AI Competitions guard against cheating should be viewed by the field as the gold standard for empirical rigor in evaluation.

- Meta-analyses across evaluations should be valued as highly in the field of AI as they are in fields such as medicine.

## 1.2. Structure of This Paper

In the remainder of this paper, we will first review the most typical structure and assumptions in traditional ML evaluation and discuss why they are insufficient for GenAI evaluations. We will examine the nature of generalization for GenAI, how this leads to specific concerns around leakage, and additionally show how goals of reproducibility and robustness in evaluation may be fundamentally at odds. We will then show how difficult the problem of leakage is even for traditional ML evaluations with some brief case studies, and look at current GenAI benchmarks that are aiming to overcome leakage and contamination. We finish with an examination of the ways that AI Competitions address these issues, discuss our recommendations, and examine alternate viewpoints. Our goal is to provide convincing support of the view that AI Competitions do indeed provide a gold standard for empirical rigor in evaluating GenAI models, and that the field should place accordingly high value and attention on their results.

## 2. Background: Revisiting Benchmarking

Traditional ML benchmarking has been founded on the idea of a *test-train split*, in which an evaluation is structured by training a model from scratch on a given portion of training data and then evaluating that trained model on a holdout set of test data (Mitchell, 1997). This conceptual structure is so fundamental to modern ML practice that it may sometimes be taken for granted. So let us take a moment to examine this basic structure and its implications.

In classical supervised ML, the most common traditional setup is to evaluate a model $f(\mathbf{x}) \to y$, with $\mathbf{x} \in \Re^d$ as feature vectors in some $d$ dimensional feature space and $y \in Y$ as a space of possible labels, such as $\{0, 1\}$ for binary classification or $(0, 1)$ for regression on probabilities. Labeled examples $(\mathbf{x}, y)$ are assumed to have come from some distribution $D$. The `training set` $D_{train}$ and `test set` $D_{test}$ are each independently and identically drawn (IID) from $D$, with only the examples in the `training set` used to fit the model $f(\mathbf{x})$ and only the examples in the `test set` used to evaluate the model (Mitchell, 1997).

The IID requirement on test-train splits is often taken as a footnote in practice, but in reality it is a cornerstone of the robustness of this setup. The reason for this is that we fundamentally wish our evaluations to be interpretable as statements on the *generalization* ability of our models: we wish to know how the model will perform on future, previously unseen data. But achieving this is harder than it may sound, because the ML models in question are often of extremely high dimensionality and thus may be prone to overfitting.

One approach for assessing generalization ability lies in the classic literature on statistical learning theory, providing generalization bounds for models based on qualities like their VC dimension and observed error during training that do not require the use of an additional holdout set (Vapnik, 1999). However, these theoretical bounds are unfortunately much too loose to be of practical value—all the more so in the age of ever larger models.

A second approach is to use additional data for evaluation. The issue here to be aware of is the classical statistical trap that correlation does not necessarily imply causation, and that trying to assess generalization ability of a model that has no specific mechanism for disambiguating correlation from causal factors may lead to wildly unreliable performance estimates. It is this issue that the IID assumption addresses. When we know that all test data is drawn IID from the same distribution as the training data, we know that all correlations that held at training time will reappear with the same characteristics at test time, and thus we can take performance on holdout test data as a reasonable estimate of generalization ability. The IID assumption has, in many ways, enabled modern ML research to advance as a field, because it forms the theoretical underpinning of all evaluations. And indeed, it is a truism that moving ML models from research to deployed production is difficult precisely because of the fact that the IID assumption often does not hold in practice (Chen et al., 2021a).

### 2.1. The Rise of Reproducible Benchmarks

One statistical shortcut that immediately became standard practice was instead of drawing a new `training set`

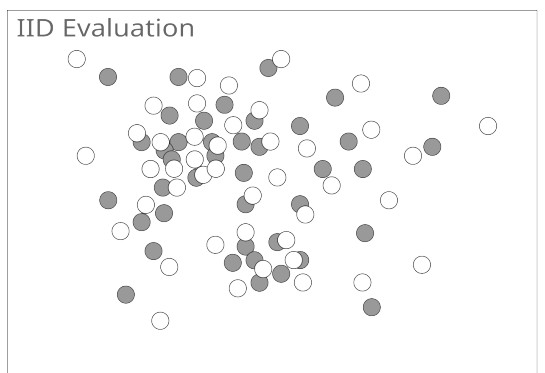 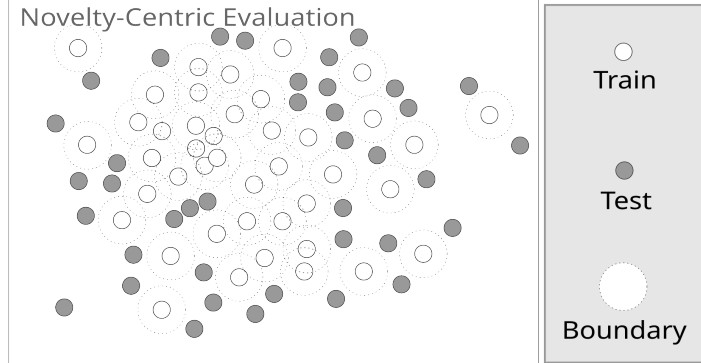

*Figure 1.* **IID Evaluations vs. Novelty-Centric Evaluations.** In the IID evaluation, left, both training and test data are drawn from the same distribution, resulting in significant overlap in examples in each set. In the novelty-centric version, right, no test example is allowed to be too similar to any given training example. We argue that the latter conceptualization more closely mirrors desired behavior for GenAI evaluations, where generalization is expected to connote the ability to respond well on totally novel inputs.

and `test set` from $D$ for every new evaluation, researchers would make one draw of each and use them as a canonical train / test set. The primary benefit of this approach (besides convenience) was that these paired test-train splits could now be used as *reproducible benchmarks*. All future researchers could replicate the exact problem setup, giving a new untrained model the same training data for training and the same test data for evaluation, allowing for full apples-to-apples comparisons. This approach was wildly successful, with canonical benchmarks such as MNIST (LeCun & Cortes, 2010) and ImageNet (Deng et al., 2009) responsible for driving incredibly rapid progress in computer vision, for instance, and benchmarks such as those of Rajpurkar et al. (2016); Marcus et al. (1993); Diemert Eustache, Betlei Artem et al. (2018) moving forward email spam classification, natural language processing, and many other domains. Websites such as the UCI Machine Learning Repository (Kelly et al., 2025) and OpenML (Vanschoren et al., 2014) among many others have been remarkably valuable to the field for this reason.

### 2.2. Surprisingly, Overfitting Was Not the Main Issue

Given that one of the fundamental concerns in evaluating models was ensuring that we could properly address generalization and avoid overfitting, it is reasonable to ask whether widespread reuse of the same standard benchmark datasets in thousands of papers might not lead to overfitting. As authors, we were deeply surprised by the work of Roelofs et al. (2019b) showing that in practice, this appears to actually not have been a problem. Recht et al. (2019) shows that the rank ordering of ImageNet models when evaluated on brand new data was remarkably consistent with the rank ordering of those models on the benchmark data, despite wide reuse. And in follow-up work, Roelofs et al. (2019a) similarly show that evaluation on public leaderboard data on Kaggle competitions was a remarkably good indicator of rank ordering on private holdout data, despite the risk of

overfitting when many thousands of teams participate in the same challenge.

## 3. Reconsidering Generalization for GenAI

As we recalled above, the IID assumption in traditional ML evaluations gives a clear conception of the idea of generalization: a model generalizes well if it accurately predicts the true-but-hidden label $y$ for labeled examples $(\mathbf{x}, y)$ drawn IID from the same fixed and stationary distribution $D$ from which the model's training data was drawn. This was a cornerstone allowing the field of ML to progress effectively by narrowing the problem and enabling tractable statistical theory. But if we reflect on broader notions of intelligence, including those first proposed in the seminal paper by Turing (1950), it is clear that this narrow-but-useful notion of generalization does not adequately reflect the deeper goals that GenAI is aiming to deliver on.

Instead, we believe **the form of generalization the field should most care about for GenAI is novelty-based generalization**—that is, generalizing well to problems and tasks that have truly never been seen before by the system in training or development.

More deeply, evaluations of reasoning and understanding that have been in our view easiest to design as tests often have the quality that solving the problems is hard (in a formal sense) or expensive, while answer verification is significantly easier or cheaper. This experience holds true in planning, solving mathematics problems, doing coding problems, solving riddles, and even formulating essays, and mirrors the fundamental quality of NP-hard problems. Once an answer for a problem is known to a subject, the ability to use that problem or very similar problems for that subject again in the future is fundamentally compromised.

In order to assess novelty-centric generalization, we need novelty-based evaluations. Informally, the goal of a novelty-

based evaluation is to ensure that no evaluation task or example is too closely similar (for some definition of *similar* and some measure of *too close*) to any instance previously known to the model or system. We illustrate the distinction between IID-based generalization and novelty-based generalization in Figure 1 visually. We can imagine a small conceptual ring around each training example and ensure that no evaluation instance crosses any of those rings.

In our view, this novelty-centric view of generalization has already been implicitly adopted by many in the field as the true aspirational goal, and influences the design of important benchmarks including the LM Arena (Chiang et al., 2024a) among others; we are simply writing this *de facto* standard down. We will now examine some of the implications.

### 3.1. The IID Assumption is Broken

While the IID assumption has often been broken in practice for traditional ML systems in real-world deployment with only modest harm, we believe that the IID assumption and the overall framework of neatly labeled examples $(\mathbf{x}, y)$ is broken beyond repair for GenAI evaluation. In particular, **the novelty-centric view of generalization strongly implies that evaluation examples should *not* be drawn from some identical distribution used for training**, but should instead be chosen or constructed with the explicit goal of avoiding high similarity with examples or data that the model has previously been exposed to.

We also note that the nature of typical GenAI models themselves leads to other ways that the IID assumption is broken. In particular, GenAI outputs are often far from independent, and instead use context of previous responses (for example, in multi-turn chat-style interfaces) to inform future responses, creating feedback loops that fully break ideas of stationarity. Finally, because the input spaces and output spaces are so vast (such as the space of all possible strings of up to a given size), the very notion of testing distributional equivalence is arguably vacuous.

### 3.2. Leakage and Contamination Are the Biggest Pitfalls

While the potential pitfall of overfitting receives strong attention, practitioners have long understood that *leakage* is an equally important and often more difficult problem in practice (Nisbet et al., 2009; Kaufman et al., 2012) Intuitively, leakage is any issue or structure in the construction of evaluation data that allows a model to "cheat" by using information that it should not have access to. In Section 4 we will look at a number of case studies on leakage and will show how hard it is to prevent leakage and how vigilant we must be even for traditional ML evaluations to avoid this pitfall. Here, we point out that leakage is an especially large problem for novelty-centric GenAI evaluations. This is because **novelty-centric GenAI evaluations have all of the leakage risks that traditional ML evaluations do, but also carry the additional burden of *novelty assurance*.**

A novelty-centric evaluation rests on assurance that a model has never before been exposed to data that is too close to the evaluation problems or tasks. While this may seem obvious, in practice it can be extremely difficult, as GenAI models like LLMs are often trained on enormous amounts of data and it can be extremely difficult to say for sure what similar data may or may not have been included. Indeed, leakage for GenAI is so important that specific forms of it have been given an additional name: *contamination* (Magar & Schwartz, 2022; Oren et al., 2023; Sainz et al., 2023; Balloccu et al., 2024). Contamination is said to occur when evaluation datasets and benchmarks appear in training data.

To help give intuition for the breadth of this issue, consider that every major LLM we have tested so far (both open and proprietary) shows extensive detailed knowledge of the contents of standard test datasets from Kaggle. Consider the remarkably strong performance on many static benchmarks by LLMs that do not seem to correlate with strong performance on other tasks (Fourrier et al., 2024; Muennighoff et al., 2023; Zheng et al., 2023). Consider the question: if a model does particularly well on qualification exam normally given to humans, is this because the model has gained strong expertise or because example examinations have appeared in its training data, and how would we be able to disambiguate? Consider the difficulty in teasing apart exactly which data sources are or are not part of an openly shared dataset such as the widely used Nectar dataset (Zhu et al., 2024), which includes the description:

> Nectar's prompts are an amalgamation of diverse sources, including lmsys-chat-1M, ShareGPT, Antropic/hh-rlhf, UltraFeedback, Evol-Instruct, and Flan. Nectar's 7 responses per prompt are primarily derived from a variety of models, namely GPT-4, GPT-3.5-turbo, GPT-3.5-turbo-instruct, LLama-2-7B-chat, and Mistral-7B-Instruct, alongside other existing datasets and models.

Together, these practical realities and considerations force leakage to the forefront of problems that must be addressed by any serious GenAI evaluation.

## 4. Leakage Case Studies

Because leakage and contamination are the most important hurdles to solve for GenAI evaluations, it is useful to study them in depth, beginning with leakage from traditional ML evaluations. Here, we draw on lessons learned surveying more than a decade of Kaggle competitions, in which a broad range of leakage issues have been identified

through intense scrutiny of a large community. Experience has shown that the risk of leakage is compounded in open ML challenge benchmarks, where teams will exploit (knowingly or unknowingly) anything that gives an advantage on the leaderboard.

Leakage can occur simply by how observations are ordered. An extreme example occurred during the SETI Breakthrough Listen competition (Siemion et al., 2021), where data was processed in order of its class label. The file timestamps were not reset, and competitors found it trivial to make predictions based on file metadata. A more subtle example occurred during the TalkingData AdTracking Fraud Detection Challenge (Yin et al., 2018), where the data was mistakenly sorted so that if multiple events were present within the same timestamp, any positive labels occurred after negative labels.

Ironically, randomization can also be a source of leakage. An example occurred during the Predict AI Model Runtime competition (Phothilimthana et al., 2023) where teams had to rank order the runtimes of 5 different subsets of data, each subset requiring a different model. Two of the buckets were randomized using the same seed, and teams discovered that using ordering of one bucket on another improved their scores.

Any data that is synthetically generated is highly prone to having artifacts that leak information. Again, in the SETI Breakthrough Listen competition, synthetically-created "ET" signals were injected into real radio telescope signals. Care was taken with normalization to ensure the averages and standard deviations of the injections matched the background signals. However, the code that created the injected signals used FP16 while the background signals were FP32. This created a minute difference in the mean and standard deviations between positive and negative samples, but enough to differentiate the classes based on this information alone.

Private evaluation data leaking to the public during an open challenge is a risk that needs to be considered. During the LANL Earthquake Prediction challenge (RL et al., 2019), for example, the test dataset was described in a research paper, including some summary statistics and a graph. A few teams discovered this and were able to utilize it to their advantage.

Space precludes a larger set of case studies, but experience from practitioners in preparing competitions shows that each AI Competition has more ways to go wrong than to go right, and that paranoia and vigilance are helpful practices. In addition to the failure modes highlighted above, other broad categories include future data leaks, the many ways metadata can leak information (e.g., the model of a medical machine being correlated to disease incidence, medical images that include a hand-drawn circle around a concern-

ing skin lesion, image aspect ratio, file size on disk, etc.), old versions of the private evaluation dataset that were not kept private, the ability of teams to reverse a synthetic data generation process or to re-assemble data that has been split up, reverse engineering data obfuscation, near duplication between training and test observations, etc. These are not hypothetical; they have all occurred in challenges created by competent, careful teams, and highlight the very real difficulty of creating leak-free competitions and benchmarks.

### 4.1. Reproducibility and Robustness in Conflict

Because of the importance of leakage and the practical difficulty in ensuring that leakage does not impact GenAI evaluations, we argue that it is simplest and safest to adopt a leakage rule of thumb that **an evaluation should be considered** `leaked` **the moment it has been shared online or sent over the wire**. Adopting this rule of thumb significantly improves our ability to trust the results of evaluations and gives them substantially more robustness. However, it also critically weakens the notion of reproducibility. It is the position of this paper that this is a fundamental tension, analogous to the Heisenberg Uncertainty Principle from quantum physics, and that we simply cannot have a published static benchmark that is robust to leakage. No matter the good intentions of the researchers, it is just too hard to avoid contamination and to broadly trust results from such a benchmark.

Instead, we must seek alternative strategies and structures to create leak-proof evaluations.

## 5. Evaluations Aiming to Avoid Leakage

Conscientious researchers have been aware of the issue of leakage in novelty-based evaluations for GenAI and have proposed new benchmarks that attempt to control or mitigate leakage through various design mechanisms. Here we review key examples, the mechanisms used to control for leakage, and briefly discuss their benefits and drawbacks.

### 5.1. Unreleased Holdout Sets

The SEAL Leaderboards (Scale AI, 2024), ARC-AGI (Chollet, 2019), FrontierMath (Glazer et al., 2024), and Humanity's Last Exam (Phan et al., 2025) benchmarks are composed of private test questions manually created by domain experts. The test sets, model responses, and evaluation runs are not published publicly to prevent leakage of test data.

Unreleased holdout sets can be effective at mitigating risks of leakage. However, they may have limitations in evaluating proprietary API-based models where test data must necessarily be sent over the internet to third party servers. While many leading AI model providers grant controls preventing logging or storage of user prompts, this still requires

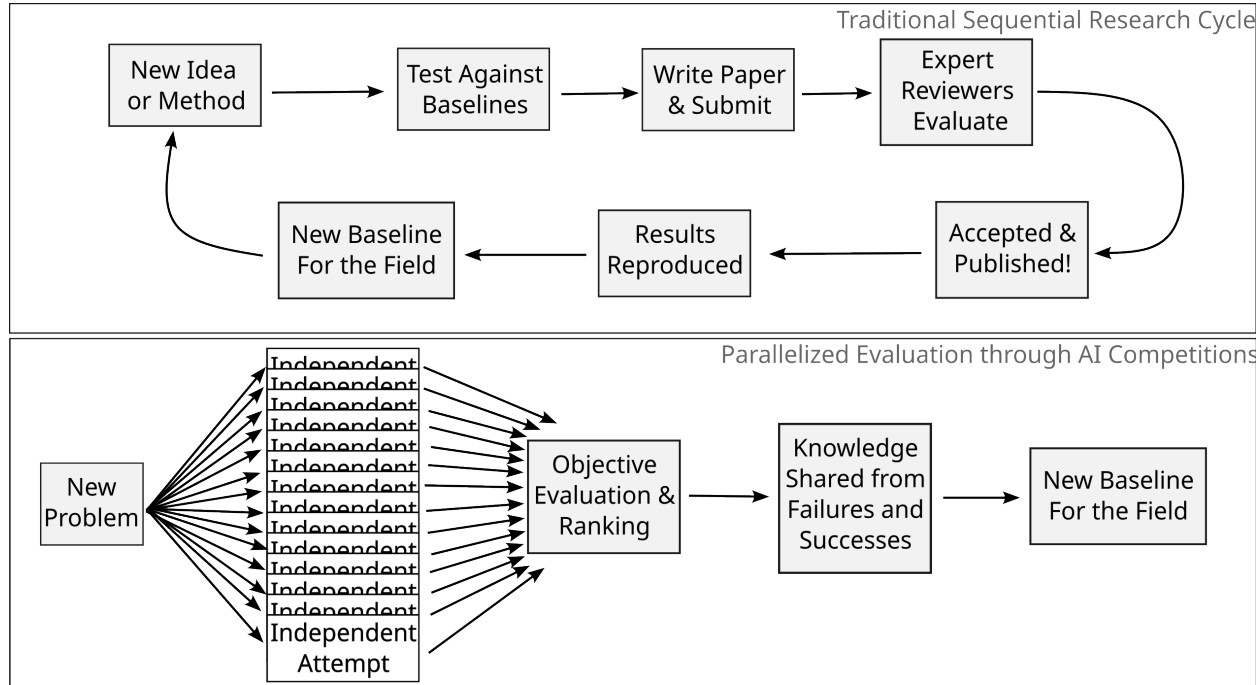

*Figure 2.* **Comparing sequential and parallelized evaluation structures.** In the traditional research structure, top, each new idea is evaluated in a linear sequence that typically requires several months for a single pass. The parallelized structure, bottom, allows hundreds or thousands of approaches to be simultaneously.

a level of trust. In particular, providers must be trusted not to change their policies, but even more importantly all researchers touching the evaluation data must be trusted to follow these practices without error.

Additionally, as holdout sets and evaluation runs must necessarily be kept private, results are not reproducible by researchers. To mitigate this, some benchmarks take a hybrid approach. For example, the FACTS Grounding Leaderboard (Jacovi et al., 2025) publishes half the test set publicly which enables partial reproducibility and better understanding of the benchmark. A model's performance can be compared between the private and public parts of test sets to identify models that may have (intentionally or not) trained on test data or leakage.

### 5.2. Dynamic Benchmarks

LiveBench (White et al., 2025), LiveCodeBench (Jain et al., 2024), and SWE-Bench (Jimenez et al., 2024) benchmarks frequently update test sets from sources that refresh naturally over time. For example, LiveBench test set questions are updated weekly from sources such as fresh news articles or papers on arXiv. By using only very recent data, benchmarks can mitigate if not eliminate the risk that such data was included in model training. LiveBench also does not release the most recently added test data so that a significant percentage of questions are always private and heldout.

Dynamic benchmarks have some advantages over unreleased holdout sets. By frequently refreshing data, older test sets can be released publicly to improve reproducibility and trust. However, using data that is publicly available on the internet—even if only very recently—does not pass our rule of thumb in 4.1 and is a potential source of leakage. Moreover, by changing the test set frequently, benchmark creators must be careful to ensure they are not "moving the goal posts." Additionally, dynamic benchmarks come with higher maintenance costs and sources of frequently refreshing data are not available in many domains and may be infeasible to collect.

### 5.3. Community Benchmarks

The LM Arena (Chiang et al., 2024a), formerly known as LMSYS Chatbot Arena, is a collection of benchmarks that draws on community votes of head-to-head match-ups between LLMs on user prompts or tasks. By outsourcing test data collection and evaluation to users at test time, the benchmark has a constant fresh source of novel test questions.

Community benchmarks are difficult to build and maintain. To evaluate many models, the number of votes required can be very large, necessitating a large and constant pool of voters. Community benchmarks also don't work for all tasks. For example, tasks that require very specialized knowledge

or which might take humans many days to verify will not scale to human rating. Community benchmarks are also necessarily biased by any sampling effects; the diversity and distribution of voters can affect results and great attention and care is required to filter out low quality, duplicate, or contaminated results.

# 6. AI Competitions as Structural Solution

AI Competitions typified by platforms like Kaggle and others offer an "embarrassingly parallel" structure to empirical evaluation shown in Figure 2 that hearkens back to the classic MapReduce structure from parallel computing (Dean & Ghemawat, 2008). In this structure, independent teams of researchers—often numbering in the thousands—each compete to solve a given problem, and in so doing create an evaluation of many different approaches in one massive parallel effort. Here we show ways that this structure offers useful benefits to the problem of GenAI evaluation at large.

## 6.1. Parallelization Improves Robustness

The risk of leakage and contamination starts as soon as an evaluation is shared publicly or evaluation data is sent across the wire. This leads to a problem: how can we fairly compare different models and systems in a valid way that ensures robustness and avoids inadvertent invalidation of results from leakage and contamination?

The parallelized structure of AI Competitions provides a useful solution to this issue. **Novelty-centric evaluations can happen simultaneously, in parallel, ensuring that each new task is indeed novel to each of the thousands of models at time of testing.** Because the independent teams each pursue different models, ideas, and approaches, this structure yields direct apples-to-apples benchmarking and a form of real-time reproduction of results.

In addition, competition platforms such as Kaggle can serve as trusted keepers of hidden test data by running isolated code competitions, where competitors submit their models to be run on an isolated, secure backend without network access. By evaluating all models securely offline, competitions platforms can guarantee no hidden test data is leaked.

Finally, competitions hosted on large community platforms offer additional non-structural characteristics which represent best practices the industry should adopt to further improve empirical rigor. Competitions encourage or often require open sharing of code, data, and experimental details, including both successes and failures. Competitors are often more motivated by the status and recognition gained from sharing valuable and insightful resources and ideas than by winning prizes. In fact, the median number of forum messages per Kaggle featured competition last year was 1,400 (data is available in the Meta Kaggle dataset (Risdal &

Bozsolik, 2022). This transparency facilitates reproduction of results, fosters trust in new baselines, and accelerates the dissemination of knowledge within the research and practitioner communities.

## 6.2. Leak-Proof Competition Structures

While preventing traditional leakage remains a challenge for competition-style evaluation as it does everywhere, competitions can be uniquely structured to mitigate this issue particularly well. Furthermore, the structure of competitions with many thousands of research teams ensures that when issues of leakage do occur, they are rapidly discovered, shared, and addressed simultaneously across all research efforts happening on the task in parallel.

We provide some examples of competitions that demonstrate the feasibility of leak-proof evaluation design. Employing strategies such as prospective ground truth, novel task generation, and post-deadline data collection, generally combined with test data that is directly inaccessible to competitors, competitions can provide a robust and reliable platform for novel evaluation of GenAI models. Overall, the main method for creating leak-proof competitions involves evaluating models based on data that does not exist at training time. These best practices should be considered and adapted as blueprints for future competition and benchmark design.

**Prospective Ground Truth** Prospective ground truth is a strategy for leakage mitigation whereby test set labels are completely unknown to the world during the active training phase of a competition.

The Critical Assessment of protein Function Annotation (CAFA) 5 challenge (Friedberg et al., 2023) is an example of a competition that uses a prospective ground truth to mitigate leakage. The competition took as its test set proteins whose sequences were known, but whose functional annotations had not yet been determined in a wet lab. Nearly two thousand participants across 1,625 independent teams therefore effectively developed models predicting the function of a set of proteins without any ground truth yet available to any human or model during an active training phase. Months later, the final evaluation was determined following a "curation phase" on the basis of newly published protein functions. This novelty makes the competition reasonably leak-proof.

**Novel Task Generation** Another approach to designing leak-proof competitions is generating novel tasks altogether in which test data doesn't resemble training data and therefore demands meaningful generalization.

The AI Mathematical Olympiad (AIMO) challenges (XTX Investments, 2024; Frieder et al., 2024), designed to motivate open progress on human-level mathematical reasoning

capabilities in GenAI systems, used this approach. In these challenges, competitors were tasked with solving national-level math challenges. Because many, if not all, AI models used by competitors were trained on internet-scale data, test-train leakage poses a significant challenge in the evaluation of their mathematical reasoning capabilities. Fresh sets of novel math problems were therefore created specifically for the competition by an international team of mathematicians, making it highly unlikely that the data has been leaked or contaminated.

**Post-Deadline Data Collection**  Post-deadline data collection is a leakage mitigation strategy used in a number of competitions which are similar to prospective ground truth competitions except rather than evaluating on newly available labels, solutions are evaluated on completely newly generated data. There are many examples of this competition design, two of which are described below.

In the WSDM Cup – Multilingual Chatbot Arena competition (Chiang et al., 2024a) hosted by LMSYS.org, participants were tasked with building solutions predicting human preferences between LLMs in head-to-head match-ups based on multilingual conversation and rating data from LM Arena. Similar to CAFA 5, this competition was designed with an active training phase followed by a data collection phase after which final models were evaluated against brand new conversations after the submission deadline in order to prevent leakage.

The Konwinski Prize (Konwinski et al., 2024) is another form of post-deadline data collection. This competition, hosted by Andy Konwinski, is a contamination-free version of SWE-Bench which evaluates LLMs on their ability to resolve real-world GitHub issues. It uses a time-based hold-out strategy in which submitted models are frozen for three months and then evaluated on fresh GitHub issues that have been collected during the intervening time.

## 7. Recommendations for the Field

As a field, we need to overhaul our standard practices to ensure that GenAI evaluations are rigorous and reliable—and that they continue to be viewed as such by the field and the broader world.

**Move away from static benchmarks and towards evergreen repeatable processes.** Due to the risks of leakage and contamination, we believe that static benchmarks should be de-emphasized in importance for GenAI evaluations. (Indeed, anecdotally we see that both researchers and practitioners are taking results from such benchmarks with ever larger grains of salt.) Instead, we need a steady renewable pipeline of novel tasks and problems, and we need to evaluate hundreds or thousands of models in parallel on each of

them so that the results are directly comparable and avoid the risks of later contamination and leakage. In this way, evaluations are best viewed as results from a point in time rather than an an immutable final conclusion.

**View the steady stream of AI Competitions as a resource for the field.** Using the pipeline of high quality AI Competitions hosted on platforms like Kaggle is one way to create a renewable pipeline. These structures already exist and are already being used to some degree in this way. However, as a field, we can do more to distill, analyze, and share findings from these competitions through meta-analyses. Indeed, while meta-analysis is a common and highly valued form of academic contribution in fields such as medicine, such papers are extremely rare in our field. We can and should change this through mechanisms that include specialized workshops, conference tracks, journal special topics, and though updated language in calls for papers emphasizing the value of meta-analyses.

Areas for fruitful meta-analysis include but are not limited to:

- **Studying Problems**: Exploring the nature and number of problems addressed via AI Competitions. Are there underrepresented problems? How well do AI Competitions connect to real-world performance on similar problems (i.e., ecological validity)?

- **Summarizing Results**: Aggregating and synthesizing trends across AI Competition results within similar domains or against similar tasks, including investigating trends (including longitudinal) in techniques, tools, models, etc. One example of this is the annual State of ML Competitions report (Carlens, 2025).

- **Improving AI Competition Design**: Meta-analyses of AI competition design including choice of objective evaluation metrics, anti-cheating and anti-contamination mechanisms, etc. to help the field develop and adopt better methodologies for AI evaluation and model building.

- **Analyses of Team and Social Collaborative-Competitive Dynamics**: Identifying patterns from the "parallelized attempts", i.e., the teams working together and against one another in competition. What characterizes successful teams and individuals across competitions?

**Adopt and improve on the anti-cheating structures from AI Competitions to improve standard practice for GenAI evaluations.** Furthermore, as a field, we can learn from the best practices that have been developed by AI Competitions. The techniques and practices that have been created to combat intentional cheating by bad actors are equally valuable

in creating evaluation structures that combat unintentional issues such as leakage and contamination that may invalidate empirical results. A cheat-proof structure is one that provides assurance to researchers that they will not accidentally cheat themselves. We also need to augment and further improve these structures, for example by creating a field-wide standard that major API-based model creators agree to follow to explicitly avoid collecting or training on data that may appear in evaluations.

## 8. Alternative Views

All position papers should consider opposing views, and ours is no exception. One reasonable alternative view is that the current state of benchmarking is proceeding well without the need for additional intervention. The many new static benchmarks appearing on platforms like Hugging Face, OpenML, and Kaggle on a near-daily basis may serve as the steady stream of novel tasks that we described as necessary for the field. While we applaud all efforts to create new benchmarks, we do fundamentally believe that static benchmarks should be considered to have been effectively invalidated once they have been published, and thus it is the *time component* of AI Competitions that provides unique additional value.

Another possible critique of AI Competitions compared to "evergreen" static benchmarks is that an artificial deadline may prohibit valuable submissions. We've found that every time we've ensembled submissions, we obtain little-to-no improvement to top ranked solutions. In other words, competitions at least on Kaggle extract (near) maximal signal from the data within the competition's constraints.

Furthermore, AI Competition hosts are strongly incentivized to design good evaluation metrics, and we observe that outcomes where solutions correlate with real-world performance are more likely. For example, in the OpenVaccine Challenge (Wayment-Steele et al., 2022), competitors improved the state-of-the-art in mRNA vaccinate degradation rate prediction by 25% within just 4 weeks, *and* the hosts further validated that the solutions generalized to much longer RNA sequences not seen as part of the competition dataset.

Another reasonable viewpoint is that current existing benchmarks that attempt to be leak-proof are sufficient. The most notable one to consider for this viewpoint are the Elo-based side-by-side rankings produced by human raters via LMSYS.org's LMArena. Having an open-loop for the community to provide an unbounded stream of new inputs and judgments is indeed appealing and is a strong step towards solving many of these issues. However, we believe there are limits to what can be achieved in terms of novelty and rigor with an anonymous crowd-based source of tasks and problems, and that AI Competitions allow for the injec-

tion of specific domain expertise and carefully crafted test cases that will fully stress test the next generation of GenAI models.

A third reasonable viewpoint is that the metaphorical ship has sailed on the value of academic evaluations for GenAI models. In this paradigm, performance on literal real-world tasks in production deployments may offer the most valid test of GenAI capabilities. In this alternative viewpoint, independent evaluations have little value and each practitioner or group should evaluate fully on their own terms. While this approach is unavoidable for highly specialized domains and applications, we do believe that there is compelling reason to continue independent evaluations of models in general, as the history of the field has shown that these forms of evaluation drive progress in the broadest and most rapid ways. Without controlled, empirical study we as a field risk losing broadly shared knowledge into *why* models perform well or poorly on certain tasks. Openly sharing this understanding is critical for unlocking paths to further progress in this rapidly advancing field.

## Acknowledgments

The authors would like to express deep gratitude to those who have played an instrumental role in building and evolving Kaggle as an AI Competitions platform that sets the gold standard today for GenAI evaluation: Anthony Goldbloom, Jeremy Howard, Ben Hamner, Jeff Moser, and many other members of the Kaggle team, past and present. We're equally grateful to the many hundreds of hosts and partners who have brought diverse and challenging problems to the AI community through Kaggle. Their contributions and experiences have led to many insights and learnings shared in this paper.

We've also benefited from discussion with members of Kaggle's Research Advisory Board: Isabelle Guyon, Jonathan Frankle, Sara Hooker, and Joelle Pineau.

Kaggle is a wholly owned subsidiary of Google, Inc. and we gratefully acknowledge Google and Alphabet for their ongoing support for this work.

Finally, we are thankful to the AI community who has participated in Kaggle Competitions and shared their knowledge openly with the world.

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
