# OpenReview forum: "Position: AI Competitions Provide the Gold Standard for Empirical Rigor in GenAI Evaluation"
_ICML.cc/2025/Position_Paper_Track — ICML 2025 Position Paper Track oral_

### Official Review · Reviewer_oyvv · 2025-03-11

**Significance:** 4
**Argument Clarity:** 3
**Rating:** 5
**Confidence:** 4

**Questions:**

To follow up on my comment above, I would like to ask how we can prevent "moving the goal post"?

**Discussion Potential:**

4

**Paper Summary:**

The paper discusses challenges and potential solutions for assessing GenAI methods. The authors start by discussing the standard train-test protocol and IID assumptions in the context of GenAI and highlight that leakage and contamination pose the largest issues. The paper then highlights dynamic benchmarks and novel-centric benchmarking as potential solutions and AI competitions as an infrastructural solution to implement this as a community.

**Position:**

Yes

**Position In Title:**

Yes

**Related Work:**

4

**Strengths And Weaknesses:**

**Strengths**
  * Topic is very relevant to the ML community
  * The paper is very straightforward
  * Careful writing and overall structure, helpful references, strong and comprehensive argumentation, and anecdotal examples make this paper a pleasure to read

**Weaknesses**
  * The paper primarily focuses on language models. It's unclear whether the position extends to other modalities (as the title suggests), e.g., vision and structured data.
  * The paper briefly mentions "moving the goal posts" (line 325; left), which seems like a potential risk when shifting towards renewing changing benchmark tasks. Comparability over time and rigorous evaluation were prime motivations for fixed benchmark sets (in contrast to using computationally extensive real-world tasks or synthetic problem generators). The need to establish new standard protocols due to leakage and contamination is very well motivated; however, a more extensive discussion of risks and limitations would strengthen the position.

**Support:**

4

---

> ### Author Rebuttal · Authors · 2025-04-01
>
> **Does the position extend beyond language models to other modalities?** Our position extends to generative models, not just language models. We’ll be happy to make this more clear in the paper.
>
> **More extensive discussion of risks & limitations** Thanks to your feedback and the feedback from other reviewers, we're prepared to offer a more in-depth discussion of the trade-offs associated with our proposal to adopt AI Competitions as the gold standard for GenAI evaluation. As you note, risks associated with "moving the goalposts" is one such challenge introduced by constantly evolving benchmark tasks in pursuit of novelty. We may leave discussion of how to address this particular challenge for future papers, but we will note that there may be parallels with data drift in MLOps to draw lessons from.
>
> Other risks and limitations we may touch in in an expanded overview may include:
>
> 1. Inability to evaluate API-based models in offline settings in AI Competitions due to risk of leakage which is impossible to prevent;
> 2. Practical challenges associated with running enough AI Competitions at scale to reflect real-world model capabilities that the industry cares about with sufficient community participation
> 3. Reluctance from model providers to provide access to unreleased models to AI Competition platforms or their users for use in evaluations;
> 4. Inherent difficulties collecting new data in prospective-ground-truth style competitions as a contamination mitigation strategy
>
> We welcome your feedback on any or all of these potential areas to flesh out further in the paper.

---

### Official Review · Reviewer_UXTh · 2025-03-16

**Significance:** 3
**Argument Clarity:** 4
**Rating:** 3
**Confidence:** 3

**Questions:**

1. L220: "...an evaluation should be considered leaked the moment it has been shared online or sent over the wire. Adopting this rule of thumb significantly improves our ability to trust the results of evaluations and gives them substantially more robustness." It isn't clear what "robustness" precisely means. Could the authors present a detailed clarification?
2. L341: "...secure backend without access to internet. By evaluating all models securely offline, competitions platforms can guarantee no hidden test data is leaked." What about proprietary models? How might they fit in with this paradigm?
3. In Fig. 2, two views are presented in the top and bottom panels. Though I understand the contrast that the authors intend to provide, is it not the case that parallel submissions to conferences/ArXiv convert the top panel into the bottom panel? If that is indeed the case, then I do not see much difference between the proposal and the existing state of evaluations.

**Discussion Potential:**

3

**Paper Summary:**

The paper takes the position that GenAI evaluations should move away from the classical strategy of using static benchmarks and instead adopt AI competitions hosted on public platforms that invite parallel submissions from many teams as the gold standard for evaluation. Case studies are presented to highlight the lacunae with the classical paradigm, as well as a description of alternative strategies used to address them, e.g., unreleased holdout tests, dynamic benchmarks, and community benchmarks. A recommendation for the field is then provided to move away from static benchmarks, conduct meta-analyses of competitions, adopt anti-cheating structures.

**Position:**

Yes

**Position In Title:**

Yes

**Related Work:**

3

**Strengths And Weaknesses:**

**Strengths:**
- The paper is well-structured and the position and arguments are clearly stated and easy to follow.
- The position is well supported with several examples and recent works.
- The Alternative Views are discussed well and do not merely provide lip service.

**Weaknesses:**
- None as such. Please see my questions instead.

**Support:**

3

---

> ### Author Rebuttal · Authors · 2025-04-01
>
> **It isn’t clear what “robustness” means**: Robustness in the context of our position paper means the degree to which a result from a GenAI evaluation can be reliably trusted to reflect a model’s capability where that capability may be reflected in unbounded input/output space. We contend that the best measure of this depends on generalizability to novel tasks which don’t resemble a model’s training data. This is why novelty-centric tasks (as opposed to IID ones as in traditional machine learning settings) and mitigations against contamination and leakage (like parallelized evaluations in a time-bound AI Competition) are critical for evaluation in GenAI.
>
> **How are proprietary models evaluated in offline settings?** This is a good question. Leakage happens when a model has access to information, data, or examples that it could potentially use to “cheat” on a given task. Sending data over the wire to a hosted proprietary model from an API provider introduces a risk of leakage. Therefore, by construing AI Competitions as the gold standard for AI evaluation, proprietary models cannot be evaluated in offline settings.
>
> We acknowledge that it’s obviously desirable and important for the field to evaluate proprietary, API-based models. It remains that benchmarks evaluating such models on GenAI tasks need to enforce novelty and mitigate cheating/contamination to the greatest extent possible, but only AI Competitions can act as the gold standard.
>
> **Parallel versus linear structure of evaluations?** On this point we would like to offer a clarification. In our view, parallel submissions to conferences/Arxiv, etc. does not constitute a parallelized evaluation structure. In the scenario you offer, just one “independent attempt” is submitted to multiple different “objective evaluation & ranking” functions (i.e., multiple venues). Our position is concerned with multiple “independent attempts” evaluated against the same “objective evaluation & ranking” function.
>
> In our position, your comparison is not apt and we maintain that the evaluation structure we propose is distinct from the current state of evaluation in the industry. Please let us know what you think and based on your thoughts we can consider refinements to our related points in the paper to make this clearer.

---

> > ### Comment · Reviewer_UXTh · 2025-04-05
> >
> > Thank you for your responses.
> >
> > Given that proprietary models currently hold outsized importance, it seems subpar that the ideal evaluation setting being proposed will not be able to accommodate them without qualification. This makes me question the generality of the position.
> >
> > Regarding parallel submissions to conferences/ArXiv, to clarify, my statement referred to parallel submission being from independent sources, not a single work submitted to multiple venues. Could you please revise your response based on this clarification?

---

> > > ### Author Response · Authors · 2025-04-09
> > >
> > > Re: proprietary models, it's a fair criticism regarding generality of the position. But we don't feel it changes the standing of AI Competitions as a gold standard *in principle*. Even if it's a gold standard which is more narrowly applicable than we wish it could be given the physical realities of leaking data over the wire to a model hosted by a 3rd party provider, it still defines an important high bar for what we consider rigorous, empirical evaluation in the era of GenAI and demands for novelty. In our view, the risk of data contamination is underappreciated and if the fact that recognizing AI Competitions as a gold standard means simultaneously doing more to acknowledge the risks of data contamination with proprietary models, then that's a healthy outcome for our industry. It's an outcome we hope can lead to better informed choices to mitigate contamination where possible even if a gold standard AI Competition is not possible in all cases.
> > >
> > > Re: parallel submissions to conferences/ArXiv, thank you for clarifying! Apologies for misunderstanding initially and maybe there's still a disconnect in how this analogy is landing. In the case you describe, though, what's being evaluated is "Is this submission worthy of acceptance to conference XYZ?" across many different submissions which is different in kind to what our position paper is focused on (and is of course how conference submissions have always worked). What we're concerning ourselves with is evaluation of GenAI models and their capabilities. That is, what is SOTA for a very specifically defined task?

---

### Official Review · Reviewer_zENR · 2025-03-19

**Significance:** 4
**Argument Clarity:** 3
**Rating:** 4
**Confidence:** 3

**Questions:**

1. My main question to the authors is how they would define "AI Competition" (see W1). Depending on this definition, some of my weaknesses might not apply as written above (or could change).
2. Do you have any evidence that competition rankings correlate with real-world performance more than static leaderboards?
3. You argue that "meta-analyses should be valued as highly in the field of AI as they are in fields such as medicine". I like this point, but it is quite short on details in the paper. Could you elaborate a bit more, e.g. what meta-analyses could look like in our field, especially when using AI Competitions?

I also have a few additional questions and comments. However, they don't fall under the category of "the response would likely change my opinion" and are thus not critical to the review process. I am still interested in the authors' response, though, or think the feedback could be helpful to strengthen the paper.

4. On a very high level, I would be interested in the authors' thoughts on how much of the GenAI evaluation issues have to do with the fact that we allow full control over the training set. I.e. in traditional ML benchmarks, $D_{train}$ was fixed (i.e. we wanted to see results on an ImageNet validation set, if trained only on the ImageNet training set). Do you think that GenAI evaluations would profit from this lens? For example, a competitive leaderboard of LLM model architecture that fixes aspects like the training set, training protocol, etc.
5. Do you think that novelty-based generalization could simply be a different task than iid generalization in the sense that both have their place but we should be transparent about which of those we are currently measuring?
6. There are a few (potential) minor typos in the paper. For example:
   - Line 34 (right): "rigorous and robust evaluation [of?] GenAI models..."
   - The use of title case for headings is inconsistent. E.g. "1.2 Structure of this paper" vs. "2.1 The Rise of Reproducible Benchmarks".
   - I think some of the citations should be text citations (e.g. \citet or \textcite) instead of bracketed citations. For example, in line 143 (left) "As authors, we were deeply surprised by the work of Roelofs et al. (2019b)".
   - Line 318 (right): Double "the" in "the the".
   - Line 318 (right): I am not familiar with the word "writ" as it is used in this sentence. Perhaps replacing it with "at" would improve readability?

**Discussion Potential:**

4

**Paper Summary:**

This position paper argues that traditional machine learning evaluation protocols, e.g. static benchmarks with publically visible IID train/test splits, are inadequate for Generative AI models, such as LLMs. This is first and foremost due to the risk of data leakage or contamination, where the same (or very similar) test data has been seen during training time. This leakage should be assumed for any data as soon as it is shared in any capacity.

Instead, AI Competitions (e.g. on Kaggle) provide a more meaningful evaluation of GenAI models, since their structure, e.g. their "anti-cheat" measurements, also reliably prevent data leakage.

## Update after Rebuttal

The authors have addressed most of my questions and concerns. I believe that the updated paper is improved and have therefore increased my score.

**Position:**

Yes

**Position In Title:**

Yes

**Related Work:**

3

**Strengths And Weaknesses:**

# Strengths

- The paper's topic is important and very timely. There are growing concerns about the meaningfulness of current empirical evaluations, especially for LLMs and other GenAI models.
- The authors have (a couple) of clear positions that they nicely summarize in Section 1.1. This not only highlights the problem (i.e. current evals are not meaningful, mostly due to leakage) but also provides an opinion on what should be done about it (i.e. AI Competitions). For me personally, the point that robustness and reproducibility can be at odds with each other was very insightful.
- The paper is well-written, easy to follow and well-presented. The added illustrations (i.e. Figure 1) help to present the author's position.
- Overall, I think the paper is helpful in sparking an important discussion. Which, in my opinion, is the main point of position papers.

# Weaknesses

Some of the logical steps feel a bit rushed or vague. This either makes them too unclear to be properly challenged, as the authors' position isn't fully defined or suggests that a key argument or supporting evidence is missing. I want to highlight this with some examples:

## W1: Missing Definition of AI Competition

In my opinion, the paper lacks a clear(er) definition of what an AI Competition is (e.g. via defining key ingredients/properties). Without this, it is hard to argue for or against AI Competitions (and thus the authors' position), if what counts as an AI Competition is not clear (and could be changed based on what is needed for the argument).
For example, from the paper, it sounds to me like, for the authors, two defining features of AI Competitions are a competitive nature and a "time component" (Section 8). However, it also sounds to me like the LM Arena is not considered an AI Competition by the authors (e.g. it is mentioned in 5.3 and also in 8.). But it has many of the key ingredients of one (both the competitive nature and time component, i.e. fresh data that is created after submitting a model).

## W2: Arguing for Novelty-Based Generalization

In my opinion, the argument that "novelty-based generalization is the most interesting generalization" is a bit rushed (I am not saying I disagree, but I believe it could be better argued). If I understand it correctly, the only real argument for this (in the paper) is "[...] if we reflect on broader notions of intelligence, [...] it is clear that this narrow-but-useful notion of [iid] generalization does not adequately reflect the deeper goals that GenAI is aiming to deliver on." (page 3). To play devil's advocate, if I build a chatbot LLM service, I mostly care about the chatbot's ability to answer my user's questions satisfactorily. If $D_{test}$ truly is the set of **all** user questions in production, I only care about the chatbot's performance on $D_{test}$. It doesn't really matter if it can answer the questions because it memorized them or if it can truly generalize to new questions.
Now, this is obviously a bit of an extreme example and argument, but I hope it highlights, why I think the argument for novelty-based generalization is a bit rushed.

## W3: Static Leaderboard vs. AI Competitions

Assuming that I understand the authors' definition of AI Competitions correctly (see W1), I think the natural "competitors" to AI Competitions are static leaderboards with a private test set, e.g. the FACTS Grounding Leaderboard (or also the ImageNet leaderboard as opposed to the ImageNet competition, since the leaderboards don't have "the time component of AI Competitions that provides unique additional value" (Section 8)).
I believe that the paper currently i) does not convincingly argue against a well-designed static leaderboard with a private test set (which in my understanding, wouldn't be an AI Competition due to the missing time component) and ii) does not fairly account for the downsides of AI Competitions (vs. static leaderboards).
i) The authors argue for novelty-based generalization and the importance of preventing data leakage. But I don't see an argument for why these two aspects can't be realized in a well-designed static leaderboard. For example, the Novel Task Generation (which is mentioned in 6.2 as part of the "unique structures of AI Competitions") could also be realized in a static leaderboard (with a private test set). As another example, Section 6.1 mentions how AI Competitions' "structures offer[s] useful benefits to the [the] problem of GenAI evaluation" (page 6). Specifically, it mentions that "platforms such as Kaggle can serve as trusted keepers of hidden test data by running isolated code competitions, where competitors submit their models to be run on an isolated secure backend [...] [to] guarantee no hidden test data is leaked." This could equally be realized for a static leaderboard with a private test set. However, it also ignores the fact that many companies developing LLM models are likely not willing to send their new and unreleased models to a third party (e.g. Kaggle) for evaluation, which leads to the next point.
ii) I believe that the paper doesn't sufficiently explore the downsides of AI Competitions (vs. static leaderboards). For example, the paper criticizes that "community benchmarks are difficult to build and maintain" (Section 5.3), but Prospective Ground Truth (which is presented as part of "competitions can be uniquely structured to mitigate" data leakage) has arguably the same problem. Gathering the ground truth data has to be difficult/expensive (by design), otherwise, it could easily be collected by submitters (and thus we would have data leakage).
Additionally, the paper mentions that we should use the "steady stream of AI Competitions as a resource for the field". I think one could argue that a downside of AI Competitions is the very fact that (due to the "time component") we only use the once. This inadvertently means that we have to create AI Competitions more frequently. So the central question becomes: Should we use the "steady stream" of novel AI Competitions (at the risk of them being of lower quality due to the need to create them more frequently) or should we re-use a (potentially more well-designed, because it has a longer shelf life) static leaderboard? This isn't only a black-and-white question, but we could consider many positions in between (e.g. how often should we refresh our benchmarks?).
There are also organizational challenges with AI Competitions, e.g. limiting the pool of submitters via submission deadlines.

# Summary

I want to emphasize that although my weaknesses section is rather long, I enjoyed reading this paper and it definitely sparked an interesting thought process (in myself), which I believe is the main point of position papers. I still believe that addressing the weaknesses above could strengthen the paper's argument.

**Support:**

3

---

> ### Author Rebuttal · Authors · 2025-04-01
>
> ## Definition of AI Competition
> Authors agree that the position paper can be strengthened by providing a clearer description of an AI Competition.
>
> We define an AI competition as a problem or task with an objective evaluation function for ranking solutions or models in which multiple parallel attempts are made during a timebound period. See Figure 2 for a visualization of the crucial parts of this definition. It has the additional properties of facilitating sharing of knowledge about success and failures openly for the world to learn from.
>
> LM Arena is an important contribution to the field, because, as we note, its design inherently elicits novel tasks in the form of user prompts. Unlike AI Competitions, however, this same design also inherently limits the nature of tasks under consideration.
>
> Our position is that AI Competitions offer a more structurally controlled and robust paradigm for evaluating GenAI models. We’d be happy to strengthen these points further in our paper in response to your feedback regarding Static Leaderboard vs. AI Competitions, too.
>
> ## Do competition rankings correlate with real-world performance?
>
> We are aware of substantial evidence that competition rankings correlate with real-world performance. One such example is documented in [Nature Machine Intelligence](https://www.nature.com/articles/s42256-022-00571-8) from the [OpenVaccine challenge run on Kaggle](https://www.kaggle.com/competitions/stanford-covid-vaccine) in 2020. The researchers were looking to improve their ability to predict degradation rates of mRNA vaccines. In four weeks, competitors improved the SOTA by 25%, and the hosts further validated that the Kaggle solutions generalized to much longer RNA sequences not seen as part of the competition dataset.
>
> Overall, the evaluation metric chosen by the competition host and the extent to which it faithfully captures their business needs is the crucial factor. As long as it measures the right thing, models will optimize correctly. Competition hosts are strongly incentivized to design a good evaluation metric, so outcomes where solutions correlate with real-world performance are more likely.
>
> ## Value of meta-analyses
>
> We’re glad this point resonates. There’s several ways we could envision meta-analyses being treated in our field that would elevate their status. Please let us know your thoughts and we can take it into consideration as we refine this piece of our position.
>
> * **Studying problems**: Exploring the nature and number of problems addressed via AI Competitions. Are there underrepresented problems? How well do AI Competitions connect to real-world performance on similar problems (i.e., ecological validity)?
> * **Summarizing results**: Aggregating and synthesizing trends across AI Competition results within similar domains or against similar tasks, including investigating trends in techniques, tools, models, etc. One example of this is the annual State of ML Competitions report, e.g., see https://mlcontests.com/state-of-machine-learning-competitions-2024
> * **Improving AI Competition design**: Meta-analyses of AI competition design including choice of objective evaluation metrics, anti-cheating and anti-contamination mechanisms, etc. to help the field develop and adopt better methodologies for AI evaluation and model building.
> * **Analyses of team and social dynamics**: Identifying patterns from the “parallelized attempts”, i.e., the teams working together and against one another in competition. What characterizes successful teams and individuals across competitions?
> * **Longitudinal analyses**: How has SOTA evolved over time? How have tools, techniques, models, etc. changed?
>
> ## GenAI evals lens
>
> This lens strikes us as slightly out-of-scope for the focus of our position. Given the current state of generative AI, it’s not practical to hold training sets or training protocols fixed across LLMs on an AI Competition task within a GenAI domain. Use of transfer learning and pretrained models has been common in machine learning competitions for many years now to the point of ubiquity today. Please feel free to clarify your question or provide more examples for us to respond to.
>
> ## Are novelty-based generalizations simply a different task to IID generalizations?
>
> Yes, we agree with your suggestion. Novelty-based generalizations represent a different type of task to IID generalizations. Our explicit position is that tasks in GenAI evaluations must be novelty-based in order to assess capabilities across a potentially unbounded input/output space. Assumptions of IID do not hold for GenAI, but still have their place in traditional machine learning settings where the data distribution is relatively stable and the goal is primarily to generalize within that distribution. We agree that it’s important to be explicit about what kind of task is being measured.
>
> ## Typos
>
> The authors thank you for your careful eye. We’ll happily address these and any other remaining issues.

---

> > ### Comment · Reviewer_zENR · 2025-04-03
> >
> > I want to thank the authors for their rebuttal and clarification. I will maintain my positive review, which is leaning towards acceptance.
> >
> > A few comments:
> > - Thanks for the definition of AI Competition. I think it could be helpful to have this explicit definition relatively early in the main text.
> > - I also think the examples of meta-analyses are useful. If the authors find it appropriate, I would encourage them to add some of these examples to the main text.
> > - I still think that the paper could be stronger in its logical arguments. For example, it argues that AI competitions are the gold standard. However, I don't really see a convincing argument that well-designed (static) leaderboards with a private test set have to be worse. On the contrary, one could argue that specifically the "time component" that defines a competition can also provide downsides (which haven't really been explored in the paper). For instance, the parallel efforts of a competition mean that there is relatively little communication between the teams. Why should I reveal my tricks before the competition deadline to competing teams? This means that the likely best solution (made by combining the strengths of multiple submissions) cannot be explored (since you suggest not reusing competitions). Similarly, the somewhat artificial deadline can prohibit interesting submissions. Now again, I am not saying I disagree with "AI Competitions are useful". However, I belive that the logical argument for "AI Competitions are the gold standard" could be stronger in the paper, especially compared to well-designed static benchmarks with private test sets.
> > - Regarding the GenAI evals lens: My question was mostly out of curiosity about what the author's expert opinion on this matter is, not a suggestion to add it to the paper. One premise of the paper was "Overfitting to benchmarks is (surprisingly) not a crucial problem for 'traditional ML' (as demonstrated in works by Recht et al. or Roelofs et al.), however, it is a more crucial problem for GenAI", right? The paper argues that this is due to the broken IID assumption in GenAI and/or unbounded input/output domains. I wondered if it mostly has to do with the fact that we allow researchers full control over the train set. For example, when comparing image classification results on ImageNet, we usually look at performance numbers of models trained on ImageNet (train set) and tested on ImageNet (valid/test set). But for image generation, we basically allow any train set (and then compare different models). Would some of the mentioned issues diminish if we set up GenAI benchmarks with fixed train sets and test sets? This would provide a better apples-to-apples comparison, e.g., when comparing model architectures. But this is just a side note, where I was curious to hear the author's opinion. It is not critical for this review.

---

> > > ### Author Response · Authors · 2025-04-09
> > >
> > > We’ll make an update to the paper to include a definition of AI Competitions early in the main text, thank you. We’ll also add examples of meta-analyses to the main text.
> > >
> > > Otherwise, thank you so much for the deeply thoughtful feedback. Overall, we take your point that we can make the thread of our logical argument stronger. Below we'll respond to some of the specific, excellent points and questions you've raised to offer additional data points, insights, etc. which we can incorporate into the main text to strengthen our case.
> > >
> > > ## "one could argue that specifically the "time component" that defines a competition can also provide downsides (which haven't really been explored in the paper)."
> > >
> > > Downsides to competitions compared to static benchmarks:
> > >
> > > * Can require more operational burden. It's a lot harder to run a competition than to simply put a dataset and paper out into the world.
> > > * Sensitive to critical errors. If halfway through a competition it's discovered that there's an error in the test data or eval or a leak or something, then it can be very difficult to fix if you're even alerted to it. If you aren't alerted to it, you can end up spending the whole competition optimizing for something useless.
> > >
> > > ## "Why should I reveal my tricks before the competition deadline to competing teams?"
> > >
> > > The median number of forum messages per Kaggle featured competition last year was 1,400 (data avail. in https://www.kaggle.com/datasets/kaggle/meta-kaggle). A significant number of these are participants helping each other. (One can peruse any competition to get a sense https://www.kaggle.com/competitions.)
> > >
> > > An recent example is from the Locating Bacterial Flagellar Motors competition. A competitor found an unlabeled public dataset, annotated 1,600 images, and the published them for everyone to use as a supplemental dataset, and he did this while the competition was running.
> > >
> > > https://www.kaggle.com/competitions/byu-locating-bacterial-flagellar-motors-2025/discussion/569921
> > >
> > > It is counterintuitive, but the community on platforms like Kaggle places a high value on collaboratively solving challenges. One can look at the CZII - CryoET Object Identification competition solution write-ups to see that most of the winning teams thanked a single user who published useful code and approaches during the competition for others to use. Community members do this because the status and recognition they gain in the community is at least as valuable to them as winning prize money.
> > >
> > > https://www.kaggle.com/competitions/czii-cryo-et-object-identification/leaderboard
> > >
> > > Separately, solutions are often published at the end (not just the winning solution and this can even be required by the competition design. This allows anyone to take "the strengths of multiple submissions" (including any negative results or dead ends) and test their effectiveness.
> > >
> > > ## "Similarly, the somewhat artificial deadline can prohibit interesting submissions."
> > >
> > > From https://www.nature.com/articles/s42256-022-00571-8
> > >
> > > "We explored whether increased accuracy in modelling could be achieved by ensembling models, that is, combining predictions from multiple models; a common feature of Kaggle competitions is that winning solutions are dissimilar enough that ensembled models frequently improve predictive ability. We found that ensembling resulted in only modest improvements (Methods), suggesting the majority of signal had been captured by the top two models."
> > >
> > > We've found this every time we've ensembled submissions. And anecdotally, we're unaware of any post-competition submissions in any competition that has made any substantial improvement above the top teams. In other words, competitions at least on Kaggle extract (near) maximal signal from the data.
> > >
> > > ## "I don't really see a convincing argument that well-designed (static) leaderboards with a private test set have to be worse."
> > >
> > > Expanding on the above, it is not clear at all that static benchmarks will have the same collaborative element as seen on Kaggle. The Kaggle competitions reward and recognize community engagement, and this incentivizes individuals to share useful information during limited time window of the competition. This collaborative aspect is emergent and not strictly a property of the design of AI Competition itself even if it is enabled by the time-based component, but it's certainly not our observation that it's a property emergent or otherwise of static benchmarks.
> > >
> > > ---
> > >
> > > Overall, our argument remains that for robust GenAI evaluation requiring novel tasks, AI competitions are the gold standard. Although, a gold standard doesn't need to mean they are the only useful standard or that there aren't limitations, we do still stand by our contention that more of the industry should increasingly look to this design for robust GenAI evaluation and knowledge from meta-analysis of results.

---

### Official Review · Reviewer_rTkE · 2025-03-24

**Significance:** 4
**Argument Clarity:** 4
**Rating:** 4
**Confidence:** 4

**Questions:**

1. The main method of creating leak-proof competition structure involves evaluating the model based on data that does not exist at the training time. Would that be a clearer summary organizing the paragraphs?
2. The authors mentioned that for reproducible benchmarks, overfitting was not the main issue. Are there more explanations why it was not the main issue?

**Discussion Potential:**

4

**Paper Summary:**

The paper argues that current evaluation methods are inadequate for GenAI evaluation, and leakage is the most significant pitfall. To address these issues, it advocates replacing static benchmarks with repeatable processes and procedures, like those used in AI competition platforms. The field of GenAI evaluation can mitigate leakage and increase robustness by adopting these established AI competition practices as the gold standard for empirical rigor.

**Position:**

Yes

**Position In Title:**

Yes

**Related Work:**

3

**Strengths And Weaknesses:**

Overall, this paper is clear, logically organized, well-motivated, and has significant contribution to the direction of model evaluation for the AI community. It discusses a core concern of the community - leakage in model evaluation for GenAI, and argues that the existing attributes of AI competition could be an effective way to mitigate concerns around leakage. The paper cites relevant literature, and draws from the fundamentals of ML - the IID assumption of training and test datasets to start the discussion. The arguments are generally well-supported by appropriate examples, such as various cases of leakage and AI competitions that took effective action to prevent leakage. It also articulates the alternative views, and counters the alternative view with a key insight - the time component of AI competition provides unique value that addresses leakage issues.

One weakness of the paper is that some arguments could be strengthened with better examples. For example, since Kaggle is part of Google (which has strong interests in winning model competition), citing Kaggle as the representative trusted keeper of hidden test data somewhat diminishes the original argument. Similarly, generating novel tasks that does not resemble training data is a good approach for leak-proof competitions, but the example of AIMO challenges specifically is diminished by the news that they might have also been compromised.

The paper may be strengthened by articulating the unique feature of a competition or an exam - parallelized within limited time scope. Both parallelization and time limits provide desirable properties to reduce leakage and cheating for model evaluation, similar to a general exam or competition.

**Support:**

3

---

> ### Author Rebuttal · Authors · 2025-04-01
>
> 1. Yes, authors agree this is a clear summary of the method to create leak-proof competitions. We can make a succinct statement along these lines in the paper.
>
> 2. The citations we offer (Recht et al. (2019), Roelofs et al. (2019b)) substantiate our claim that overfitting is not the largest source of issues with regard to generalization capabilities based on evidence from widely reused benchmarks (ImageNet, competitions with many thousands of participants). Recht et al. (2019) demonstrates, rather, that the issue stems from failure to generalize to novel samples. From this we contend, particularly for GenAI applications where performance on novel tasks is crucial, prior exposure to similar data (contamination) and access to information a model shouldn’t have (leakage) pose greater issues.

---

### Decision · Program_Chairs · 2025-04-30

**Decision:**

Accept (oral)

**Comment:**

This paper hits all the criteria well for the position paper track, and is especially timeline. I personally appreciated that, beyond identifying issues with current evaluation practices, the paper takes a constructive position, arguing for why AI Competitions are a serious solution the community should pursue and develop further. The reviewers clearly appreciated this paper for similar reasons. I expect this paper will generate discussion and be informative to the large and growing community of researchers thinking about evaluation of GenAI models.

The weaknesses identified were mostly addressed by the authors during the rebuttal, and the authors are encouraged to incorporate feedback from that discussion into the camera ready paper.